# Progranulin A Promotes Compensatory Hepatocyte Proliferation via HGF/c-Met Signaling after Partial Hepatectomy in Zebrafish

**DOI:** 10.3390/ijms222011217

**Published:** 2021-10-18

**Authors:** Keng-Yu Chiang, Ya-Wen Li, Yen-Hsing Li, Shin-Jie Huang, Chih-Lu Wu, Hong-Yi Gong, Jen-Leih Wu

**Affiliations:** 1Department of Life Science, National Taiwan University, Taipei 10617, Taiwan; cky0913@gate.sinica.edu.tw; 2Institute of Cellular and Organismic Biology, Academia Sinica, Taipei 11529, Taiwan; yawenli0824@gmail.com (Y.-W.L.); li4094@gmail.com (Y.-H.L.); shinjiehuang1@gmail.com (S.-J.H.); 3Department of Chemistry and Biochemistry, National Chung Cheng University, Chiayi 62145, Taiwan; chejwu@ccu.edu.tw; 4Department of Aquaculture, National Taiwan Ocean University, Keelung 20224, Taiwan; hygong@mail.ntou.edu.tw; 5Center of Excellence for the Oceans, National Taiwan Ocean University, Keelung 20224, Taiwan; 6College of Life Sciences, National Taiwan Ocean University, Keelung 20224, Taiwan

**Keywords:** progranulin, HGF, c-met, liver regeneration, partial hepatectomy, zebrafish, intraperitoneal injection, Vivo-Morpholino, cell cycle, cell proliferation

## Abstract

Compensatory hepatocyte proliferation and other liver regenerative processes are activated to sustain normal physiological function after liver injury. A major mitogen for liver regeneration is hepatocyte growth factor (HGF), and a previous study indicated that progranulin could modulate c-met, the receptor for HGF, to initiate hepatic outgrowth from hepatoblasts during embryonic development. However, a role for progranulin in compensatory hepatocyte proliferation has not been shown previously. Therefore, this study was undertaken to clarify whether progranulin plays a regulatory role during liver regeneration. To this end, we established a partial hepatectomy regeneration model in adult zebrafish that express a liver-specific fluorescent reporter. Using this model, we found that loss of progranulin A (GrnA) function by intraperitoneal-injection of a Vivo-Morpholino impaired and delayed liver regeneration after partial hepatectomy. Furthermore, transcriptome analysis and confirmatory quantitative real-time PCR suggested that cell cycle progression and cell proliferation was not as active in the morphants as controls, which may have been the result of comparative downregulation of the HGF/c-met axis by 36 h after partial hepatectomy. Finally, liver-specific overexpression of GrnA in transgenic zebrafish caused more abundant cell proliferation after partial hepatectomy compared to wild types. Thus, we conclude that GrnA positively regulates HGF/c-met signaling to promote hepatocyte proliferation during liver regeneration.

## 1. Introduction

Liver is the largest internal organ in the human body and it plays a major role in maintaining physiological homeostasis by performing many functions, including glycogen storage [1], detoxification [2], and bile secretion [3]. To ensure proper liver function, there are two major growth phases for the tissue; the first is during embryonic development [4], and the second is during liver regeneration [5]. When the liver is injured by virus infection, toxins, autoimmune disease, partial hepatectomy, or a liver tumor, it can rapidly and effectively repair itself [6]. However, uncontrollable excessive hepatic parenchymal cell proliferation leads to the liver cancer, also known as hepatocellular carcinoma, which is notorious for its high incidence and mortality rates [7]. To treat liver cancer at early stages, one efficient therapy is surgical resection (also called partial hepatectomy) [8]. However, in clinical practice, the regenerative capacity of individuals with chronic liver injury is insufficient, limiting the use of this therapeutic option. Chronic liver injury leads to cirrhosis, which is characterized by the replacement of functional epithelial tissue with non-functional connective tissue [9]. Thus, the therapeutic options for individuals with chronic liver injury may be enhanced by a further understanding of the mechanisms controlling regenerative capacity, including those involving growth factors and cytokines [10]. There are several processes that must occur for successful liver regeneration to take place, including cellular activation, proliferation, differentiation, and survival [11]. These processes are regulated by a number of cytokines and growth factors that are either expressed at the site of liver injury or transmitted via the bloodstream [5]. Cooperative signals from these cytokines and growth factors allow resident hepatocytes to pass through cell-cycle checkpoints, re-enter G1 phase, and further perform DNA synthesis and cellular proliferation [12,13].

Previous studies demonstrated that hepatocyte growth factor (HGF) is a key mitogen controlling hepatocyte proliferation [14,15]. HGF acts through a specific tyrosine kinase receptor called c-Met, which is also known to be necessary for liver regeneration in mammals [16,17,18]. Overexpression of full-length HGF or a truncated variant in the liver of transgenic mice causes increased hepatocyte proliferation and accelerated liver regeneration after partial hepatectomy [19]. Moreover, the level of HGF is upregulated in non-parenchymal cells of the liver, such as Kupffer cells, in addition to its upregulation in extra-hepatic organs after partial hepatectomy [20]. Overall, these studies showed that HGF/c-Met signaling is vital to liver regeneration after injury, and it cannot be compensated for by other growth factors [10].

Progranulin is an epithelial tissue growth factor (also known as proepithelin, acrogranin, and PC-cell-derived growth factor) that has been implicated in development, inflammation response, wound healing, and the progression of many cancers [21]. In our previous study, we showed that progranulin A (GrnA), an orthologue of mammalian PGRN, transcriptionally modulates c-met expression in embryonic zebrafish to regulate hepatic outgrowth via MAPK signaling [22]. However, the embryonic liver development begins from specification of pluripotent stem cells to hepatoblasts and proceeds to proliferation, then differentiates into mature hepatocytes and bile epithelial cells [4]. Liver regeneration by partial hepatectomy is the acute liver injury, the liver mass is restored majorly by rapid proliferating pre-existing hepatocytes [5]. The two processes in the liver are quite distinct but not totally exclusive [23]. An earlier study revealed that the transcriptional gene regulation between liver regeneration after hepatectomy and embryonic liver development was dramatically dissimilar [24]. Nevertheless, the regulatory roles of progranulin in liver regeneration are still elusive. According to previous study, progranulin was shown to regulate retinal regeneration through the c-met receptor [25], and recent studies have also revealed that progranulin is upregulated by IL-6 to promote cholangiocarcinoma, a type of liver cancer with high mortality, although progranulin was previously regarded as a promising therapeutic target [26,27]. Cholangiocarcinoma treatment sometimes includes partial hepatectomy, after which compensatory hepatocyte regeneration will occur. On the basis of this background, we were interested in exploring whether there is a regulatory relationship between progranulin and compensatory hepatocyte regeneration.

In this study, we specifically sought to clarify whether GrnA contributes to liver regeneration after partial hepatectomy. To address is question, we selected zebrafish as a model system because a number of recent studies have used it to study regeneration of many cell types and tissues, including neurons, heart, skin, fin, and liver [28,29,30,31,32]. Together, our results reveal a regulatory relationship between GrnA and the HGF/c-met axis during liver regeneration.

## 2. Results

### 2.1. Zebrafish Progranulin A Is Involved in Liver Regeneration of Hepatectomized Zebrafish

In a previous study, the growth factor, progranulin, was reported to participate in embryonic hepatic growth and hepatic cholangiocarcinoma growth progression. To study the function of GrnA in zebrafish liver regeneration, we used a transgenic line that expresses a liver-specific green fluorescent reporter, *Tg(fabp10:EGFP)*, and performed partial hepatectomy (PHx) to stimulate liver regeneration (Figure 1A). The zebrafish liver regenerated after injury or PHx typically within 168 h (Figure 1B–E).

Using this model, we analyzed the gene expression levels in the regenerating zebrafish liver. The cell cycle- and cell proliferation-related gene expression response of *Tg(fabp10:EGFP)* after PHx was quantified in triplicate by collecting five tips of the remaining liver ventral lobe in each sample and comparing to uncut liver. The results showed that liver *ccnd1* expression levels increased by 2.2-fold at 24 h and peaked at 4.4-fold higher at 36 h after PHx (Figure 1F). Liver *ccne* expression level was upregulated by 5.3-fold, 2.4-fold, 0.6-fold, and 2.8-fold higher at 24 h, 36 h, 48 h, and 72 h after PHx, respectively (Figure 1G). Furthermore, liver *ccna2* expression level after PHx was upregulated by 14.1-fold at 36 h and eightfold at 48 h after PHx (Figure 1H). Liver *myc* expression levels peaked at 36 h after PHx, with 8.3-fold upregulation over uncut liver (Figure 1I). The expression levels of *foxm1* increased by 3.3-fold, 4.4-fold, and 3.7-fold at 24 h, 48 h, and 72 h after PHx, respectively; the expression peaked at 9.4-fold upregulation at 36 h after PHx (Figure 1J). Liver *jun* expression level was upregulated at 24 h after PHx by 1.6-fold (Figure 1K). The results also indicated that *c-met* expression levels were upregulated at 36 h after PHx by 6.1-fold (Figure 1L). Liver *GrnA* expression levels peaked at 36 h after PHx with 12.9-fold upregulation. In summary, the cell cycle- and cell proliferation-related genes were activated after PHx in *Tg(fabp10:EGFP)*. Notably, the autocrine growth factor, *GrnA*, expression peaked at 36 h, suggesting it might participate in liver regeneration after PHx.

### 2.2. Knockdown of GrnA Delayed Liver Regeneration after Partial Hepatectomy

We next aimed to study the function of zebrafish GrnA in liver regeneration after PHx. A gene-specific antisense oligonucleotide, Vivo-Morpholino, was designed to knock down endogenous GrnA expression. GrnA vivo-MO was designed against the exon-intron junction between exon 6 and intron 6 of the *GrnA* gene and had the sequence 5′-GGCTTAACTCTGCTCAATACCTTTT-3′ (Appendix A). Three-month-old adult *Tg(fabp10:EGFP)* zebrafish were administered intraperitoneal injections (IP-injections) of 5 μL GrnA vivo-MO at concentrations of 0.5, 0.25, and 0.125 mM (PBS as a control) using MICROLITERTM Syringes (HAMILTON), and the fish survival rate (n = 20 per group) was assessed every 12 h after injection. At 24 h after injection, the survival rate of the 0.5 mM group was 5%, and the 0.25 mM group survival rate was 45%. Fish treated with 0.125 mM exhibited the highest survival rate of all groups at 95% at 24 h post-injection (Appendix A). Therefore, 0.125 mM GrnA vivo-MO was selected for knockdown of GrnA expression in *Tg(fabp10:EGFP)*. The injections were made every 24 h because the suppressive effect (0.39-fold expression) was observed at 24 h, but the expression recovered at 48 h post-injection (Appendix A). GrnA vivo-MO injections every 24 h could maintain GrnA expression at levels between 0.46-fold and 0.57-fold until 96 h post-injection. Notably, two control GrnA vivo-MOs, namely, Ctrl vivo-MO (contains five base-pair mismatches for short vivo-MO) and GrnA vivo-MO sense, failed to suppress the GrnA expression (Appendix A).

To ensure the GrnA gene expression was decreased before PHx, we pretreated *Tg(fabp10:EGFP)* zebrafish with 5 μL GrnA vivo-MO (0.125 mM) by IP injection 24 h before surgery. The GrnA expression in *Tg(fabp10:EGFP)* controls (treated with Ctrl-MO) was activated after PHx and peaked at 6.2-fold upregulation 36 h post-injection. In contrast, GrnA expression in the GrnA morphant was blocked significantly at every time point after PHx (Figure 2A). The GrnA protein expression of GrnA morphants was also significantly reduced compared with *Tg(fabp10:EGFP)* control, according to Western blotting at 48 h after PHx (Figure 2B). For all groups, the surgical procedure (n = 80 fish in each group) had a postoperative survival rate of approximately 91.25%, and a survival rate of 86.25% after overnight recovery. The liver was marked with EGFP expression driven by a liver-specific promoter, and the area of liver ventral lobe was calculated using ImageJ. After normalizing the liver area to the area before PHx, we found the control group livers could recover to their original size (101.5% ± 23.63%) within 7 days after PHx. However, the GrnA morphant had a slower recovery, and 7 days after PHx had only partially regenerated livers (58.06% ± 5.7%) (Figure 2C,D). As shown in Figure 2E,F, the number of PCNA-positive hepatocytes in the control group was increased up to 7 days after PHx, peaking at 2 days after PHx. Conversely, the number of PCNA-positive hepatocytes in GrnA morphants was dramatically decreased compared to the control group at 2 and 4 days after PHx; it was slightly higher at 7 days than 2 or 4 days after PHx. Therefore, the results indicated that hepatectomized zebrafish with knockdown of GrnA expression had impaired liver regeneration with correspondingly low levels of hepatocyte proliferation.

### 2.3. Transcriptome Analysis

After finding that GrnA knockdown can limit regeneration, we characterized the transcriptional alterations during liver regeneration in control fish and those with GrnA knockdown using RNA-Seq and IPA. Canonical pathway analysis was performed on Ctrl-MO-injected controls and GrnA vivo-Mo-treated fish at 24 and 36 h after PHx, compared with 0 h after PHx. The comparisons of canonical pathways revealed significantly affected pathways related to the cell cycle, including chromosome replication, cyclins, and cell cycle regulation, among others. The heatmap for canonical pathway analysis in GrnA morphants at 24 h and 36 h after PHx showed lower activation z-scores compared with the control group (Figure 3A). We then used the IPA downstream effector analysis to predict increases or decreases in downstream biological activities and functions. The comparison analysis of biofunctions revealed that cell survival, cell viability, and cell progression were activated at 24 h and 36 h after PHx. However, when comparing between Ctrl-MO and GrnA vivo-MO treatments after PHx, we found that the activation z-scores of diseases and biofunctions in GrnA morphants at 24 h and 36 h after PHx were lower than the control group (Figure 3B). To further predict the downstream effectors of GrnA knockdown during liver regeneration, we compared the biofunctions and gene network analysis results for GrnA morphants at 24 h after PHx with those for controls. The biofunction analyses indicated enrichments in genes related to organismal development, tissue morphology, and cell death and survival with GrnA knockdown (Figure 3C). The gene network analysis suggested potentially important roles of upregulated genes, including *MMP9*, *SAT1*, *PTGS2*, and *TOB1*, as well as downregulated genes, including *HGF*, *MAP2K7*, *LRAT*, and *CCNE1*, when GrnA was knocked down during liver regeneration (Figure 3D). Overall, the RNA-seq analysis results suggested that GrnA knockdown delays liver regeneration possibly by diminishing activation of cell cycle processes. Among the predicted downstream genes, downregulated HGF was predicted as a key mediator of GrnA knockdown effects on liver regeneration.

### 2.4. GrnA Morphants Exhibited Impaired Liver Regeneration with Suppressed Cell Cycle and Cell Proliferation

According to transcriptomic analysis results, activation of the cell cycle and cell proliferation is a major event in liver regeneration. We therefore measured the proportions of hepatocytes in different stages of the cell cycle in GrnA morphants after PHx. Hepatocytes were isolated at 0 h and 24 h after PHx and stained with propidium iodide (PI) for cell cycle analysis by flow cytometry (Figure 4A). The proportions of cells at each stage of the cell cycle were not different between control and GrnA morphants at 0 h after PHx (Figure 4B); however, GrnA morphants showed increased percentages of cells at G0-G1 phase (78.9%) and decreased cell proportions in the S phase (17.4%) compared to the G0-G1 phase (49.5%) and the S phase (44.6%) of controls (Figure 4C). This result suggested that the G1/S transition might not occur, and cell proliferation might be suppressed in GrnA morphants. Therefore, G1/S-associated cyclin genes, including *ccnd1* and *ccne*, were selected for examination. In addition, *ccna2*, which peaks in middle of S phase, was also examined. The *ccnd1* expression levels in GrnA morphants were significantly decreased compared with control group at 36 and 48 h after PHx, and their expression peaked late at 72 h after PHx (Figure 4D). The *ccne* expression levels in the control group at 24 h and 36 h after PHx were dramatically activated, with 23.5-fold and 11.5-fold upregulation compared to 0 h after PHx, respectively. However, the expression pattern in GrnA morphants was flat, and the difference was especially obvious at 24 h and 36 h after PHx (Figure 4E). The *ccna2* expression levels of the control group peaked at 37.7-fold higher than 0 h after PHx; however, expression levels in the GrnA morphant were not highly elevated at 24 h and 36 h after PHx (Figure 4F). We also examined *myc* expression, a transcription factor involved in cell proliferation, and the results showed that the increasing *myc* expression levels ranged from 8.4-fold to 16.1-fold higher at 12 h to 72 h in comparison with 0 h after PHx. Conversely, *myc* expression in the GrnA morphant was not elevated as much as in the control group at 36 h after PHx (Figure 4G). These results indicate that the GrnA morphant not only maintained G1/S cell cycle arrest by a lack of *ccnd1*, *ccne*, and *ccna2* upregulation, but it also had low cell proliferation according to lack of *myc* activation.

### 2.5. Knockdown of GrnA Prevented Signaling through the HGF/C-Met Axis after Partial Hepatectomy

The transcriptome analysis suggested that a regulatory relationship may exist between GrnA and HGF/c-met axis after PHx. To clarify this mechanism, we examined the expression levels of *HGFa* and *HGFb*, which are paralogous genes in zebrafish, as well as *c-met*. *HGFa* expression showed peaks at 12 h and 48 h after PHx, with 29.3-fold and 11.8-fold upregulation compared to 0 h after PHx. The peak expression levels of *HGFa* were much lower in GrnA morphants after PHx (Figure 5A). The *HGFb* expression pattern differed from *HGFa* after PHx, as it peaked with an eightfold higher level at 12 h after PHx and gradually declined to 1.9-fold elevation at 12 h after PHx, compared to 0 h after PHx. Similar to the *HGFa* result in the GrnA morphant, the *HGFb* upregulation was significantly lower at 12 h and 24 h after PHx than controls; however, a slightly increase was observed at 48 h after PHx in comparison with controls (Figure 5B). The HGF-specific receptor, *c-met*, also showed expression levels from 24 h to 72 h after PHx that were lower in GrnA morphants than controls (Figure 5C). Next, we examined the protein levels of HGF and c-met in GrnA morphants at 48 h after PHx by Western blotting. The results showed that c-met and HGF protein expression levels were lower than the control group, at 0.25-fold and 0.86-fold compared to controls, respectively (Figure 5D). We also examined the expression of c-met upstream genes, *MMP2*, *MIF*, and *YBX1*, and downstream genes, *Sub1*, *Rac1*, *rps6bk1*, and *FN1*, at 36 h after PHx. The results indicated all of these genes were significantly lower in GrnA morphants than controls (Figure 5E). Together, the results suggest a generally lower level of signaling through the HGF/c-met axis in morphants compared to controls, which would be expected to lead to impaired liver regeneration.

### 2.6. Liver-Specific Expression of GrnA in Transgenic Zebrafish Promoted Hepatocyte Proliferation after Partial Hepatectomy

To understand whether liver regeneration would be promoted by GrnA, we generated a transgenic fish with liver-specific overexpression of GrnA, *Tg(fabp10:GrnA/HcRed)*. The line was established by utilizing two *Tol2* transposon-mediated transgenesis vectors. The first vector contained a zebrafish liver-specific *fabp10* promoter/enhancer that drives expression of a tTA. The second vector contained an *HcRed* reporter gene that bi-directionally expresses GrnA in zebrafish. The genes are under the control of a *TRE* fused with *CMV* promoter, which is activated in liver by the tTA transactivator expressed by the first vector (Appendix A). The liver-specific overexpression of GrnA was verified in 7 h post-fertilization larvae by whole-mount in situ hybridization (Appendix A). The expression was also probed in livers of 6-month-old adult zebrafish (n = 5), which had a *GrnA* expression level 73.4-fold higher that wild-type zebrafish (Appendix A). The red fluorescent protein that was expressed by the HcRed/GrnA bi-directional vector was distributed homogeneously throughout the liver dorsal and ventral lobes in *Tg(fabp10:GrnA/HcRed)* (Appendix A). After validation of the transgenic, it was used for one-half ventral lobe PHx (Appendix A).

We then probed the effects on gene expression of growth factors, cell cycle-related proteins, and cell proliferation factors after PHx. The *GrnA* and *HGF/c-met* gene expression was also examined in the transgenic zebrafish after PHx. *GrnA*, *HGFa*, *HGFb*, and *c-met* were highly upregulated compared to controls, with 23-fold, 11.9-fold, 15.1-fold, and 4.7-fold differences, respectively, at 36 h after PHx, (Figure 6A). The cell cycle regulatory genes, *ccnd1* and *ccne*, were also more highly upregulated, with sevenfold and 12.8-fold differences compared to the control group at 36 h after PHx, respectively. Interestingly, *ccna2* expression was not higher in the transgenic fish compared to controls. The cell proliferation regulatory gene, *myc*, was upregulated, as high as 24.6-fold over control (Figure 6B). The liver regeneration was reflected by the numbers of PCNA-positive hepatocytes after PHx. Proliferating hepatocytes in liver-specific GrnA-overexpressing transgenic zebrafish were strongly increased at 2.7-fold over the number in the control group at 48 h after PHx (Figure 6C). On the basis of these results, we concluded that liver-specific overexpression of GrnA promotes hepatocyte proliferation after PHx, probably via upregulation of the HGF/c-met axis.

## 3. Discussion

The mechanisms of liver regeneration have been well studied over the past three decades. Many growth factors, cytokines, and transcription factors are now known to participate in regenerating the liver to sustain the physiological functions [10]. A few of these factors have been functionally characterized and identified as being involved in signaling crosstalk. For example, in addition to its critical role not in early embryonic hepatic development [33], the HGF/c-met axis was found to be a key factor in liver regeneration after injury [5]. Notably, HGF/c-Met is necessary for liver regeneration, and its role cannot be replaced by other growth factor signaling mechanisms [10]. In previous studies, it was revealed that progranulin transcriptionally regulates c-met in zebrafish during early hepatic development [22] and in mice during retinal regeneration [25]. However, progranulin function in hepatocellular carcinoma is mediated through the mTOR pathway [34], and in cholangiocarcinoma, it involves the IL-6/progranulin/Akt axis [26,27,35]. Cholangiocarcinoma is a type of liver cancer with high mortality, and one of the most efficient therapies for this cancer is partial hepatectomy [36,37]. On the basis of this previous work, we wanted to assess whether progranulin is involved in liver regeneration.

Zebrafish is an animal model with several advantages, including its fast growth and ease of observing liver regeneration. In our previous study, a liver-specific transgenic zebrafish *Tg(fabp10:EGFP)* was established to study development and disease longitudinally in the same animals [38,39]. Therefore, we took advantage of this powerful model and performed survival surgery on the transgenic zebrafish to study liver regeneration.

First, we ensured the initiation of liver regeneration by monitoring the activation of the cell cycle and cell proliferation after partial hepatectomy. There are two lobes in zebrafish liver, the dorsal lobe (≈55.6% of liver) and ventral lobe (≈44.4% of liver), which are named on the basis of their relative positions in the body cavity (Figure 1A). We performed a one-half partial hepatectomy on the ventral lobe because whole ventral lobe resection leads to poor regeneration (data not shown). The one-half resected ventral lobe was regenerated quickly in wild-type fish, recovering its original size within 7 days. During the process, cell cycle regulatory genes, including *ccne*, *ccnd1*, and *ccna2*, were upregulated. The induction of hepatocyte proliferation-related genes, including *myc*, *foxm1*, and *jun*, was also confirmed. These results were indicative of liver regeneration in hepatectomized wild-type zebrafish [32]. *C-met* expression was activated, and the expression pattern was synchronized with the cell cycle regulatory genes, *ccnd1* and *ccne* [14]. Notably, liver expression of *GrnA* was also stimulated and peaked at 36 h after PHx. Interestingly, interleukin 6 (IL-6), a pro-inflammatory cytokine known to function upstream of progranulin in cancer progression [34], plays a critical role in early initiation of liver regeneration [5]. Thus, IL-6 might be an upstream stimulator of progranulin upon liver injury as well. Together, these results indicated that aside from the role GrnA plays in embryonic hepatic outgrowth [22], it is also active in compensatory hepatocyte proliferation after injury.

To clarify the regulatory role of GrnA during liver regeneration, we used a reverse genetics approach to study the molecular mechanism. The loss-of-function GrnA morphant was established by intraperitoneal injection of Vivo-Morpholino in adult zebrafish [40]. The Ctrl-MO was designed based on a 5 base-pair mismatch of GrnA vivo-MO and was injected into the control group [41]. The results revealed that GrnA vivo-MO could inhibit GrnA expression efficiently and specifically. When GrnA expression was suppressed, the recovery capability of liver was attenuated; however, at late experimental time points, the recovery capability was restored, according to the increased PCNA-positive hepatocytes seen at 7 days after PHx. This result suggested that impaired GrnA expression can be compensated for by other processes. For instance, IGF-1 was reported as a co-induced growth factor by growth hormones [42] that modulates the downstream PI3K/AKT pathway in liver regeneration [43].

Cell cycle processes are key for hepatocyte proliferation during liver regeneration. From our canonical pathway analysis of transcriptomic data, we discovered that cyclins and cell cycle regulators, chromosomal replication factors, and cell cycle checkpoint proteins were not highly expressed after partial hepatectomy in GrnA knockdown zebrafish. Further prediction of downstream effectors revealed that cell viability, cell cycle progression, and G1/S transition were likely to be less active under GrnA knockdown. Moreover, gene network analysis showed that HGF was downregulated by GrnA knockdown, which is also an important regulator of the cell cycle during liver regeneration [10,33]. However, another important gene from network analysis, MMP9, is upregulated under GrnA knockdown at 24 h PHx. Since loss of MMP9 delays liver regeneration [34], we speculate that MMP9 may not act downstream of GrnA in liver regeneration.

According to the transcriptomic analysis, activation of the cell cycle and promotion of cell proliferation were attenuated in the GrnA morphants. Therefore, we assessed the proportions of cells at each stage of the cell cycle and gene expression levels of cell cycle factors in GrnA morphants. The cell cycle progression was evaluated with PI staining. The proportions of hepatocytes at various cell cycle stages did not differ between controls and GrnA morphants at 0 h after PHx. At 24 h after PHx, hepatocytes in the control group had accumulated in S phase; however, the proportions of G0-G1 and S phase cells were similar from 0 h to 24 h after PHx. In addition, the proportion of G2-M cells did not increase significantly. Combined with the results showing lower levels of *ccn1*, *ccne*, and *ccna2* expression at 36 h after PHx, these data indicate GrnA morphants are defective in cell proliferation upon partial hepatectomy. Additionally, the gene and protein expression of HGF paralogues [44], *hgfa* and *hgfb*, and *c-met* in the GrnA morphant after partial hepatectomy were lower than in controls from 12–72 h after PHx. These data suggest that knockdown of GrnA may cause impaired liver regeneration by attenuating the HGF/c-met axis. According to previous studies, c-met plays a critical role modulating not only G1/S phase [14] but also G2/M progression [15] during liver regeneration. Notably, suppression of the cell cycle regulatory genes occurred mostly before 36 h after PHx in the GrnA morphants. Thereafter, the fish showed upregulation of cell cycle regulators.

The gain-of-function liver-specific overexpression of GrnA in transgenic zebrafish caused abundant hepatocyte proliferation with highly upregulated HGF/c-met axis signaling after partial hepatectomy. Interestingly, *myc* expression was activated dramatically in the transgenic line at 36 h after PHx. A previous study demonstrated that progranulin can stimulate phosphorylation of myc through the ERK1/2 or PI3K pathways in breast cancer cells [45]. However, the direct or indirect regulatory mechanism between progranulin and HGF in the context of zebrafish liver regeneration is still unclear. Previous evidence indicates that HGF and c-met can be regulated by AP-1 or EGFR [46,47]. The effects of progranulin on transcriptional upregulation of c-met were also demonstrated in previous studies [22,48]. Therefore, direct and indirect regulatory mechanisms between progranulin and c-met may be assessed in future studies on liver regeneration. Intriguingly, the recovery period for full liver regeneration in the GrnA-overexpressing line was still not less than 7 days (data not shown). The timing may therefore be set by another regulator, such as TGFβ [5], which may temper over-abundant cell proliferation during liver regeneration to prevent uncontrolled growth and potential carcinogenesis.

Overall, in this study, we show that the autocrine growth factor, progranulin, is highly expressed in liver regeneration, and it likely upregulates the HGF/c-met axis to promote compensatory hepatocyte proliferation during liver regeneration after partial hepatectomy.

## 4. Materials and Methods

### 4.1. Zebrafish

The wild-type (AB) zebrafish (Danio rerio), *Tg(fabp10:EGFP)*, and *Tg(fabp10:GrnA/HcRed)* were used for experiments at 6 months of age. Approximately 30 fish of mixed sex were kept in recirculation systems (≈28 °C water, pH 7.5–8) in 10 L aquaria under a 14:10 h light/dark cycle; fish were fed twice daily [49]. Tricaine methanesulfonate (MS-222) (0.125 mg/mL) (Sigma-Aldrich Inc., St. Louis, MO, USA) was used as a zebrafish anesthetic agent, and overdose of MS-222 (500 mg/L) was used for zebrafish euthanasia in all experiments. The animal protocols used in this study were approved by the Institutional Animal Care and Use Committee of Academia Sinica (AS IACUC), Taiwan.

### 4.2. Morpholino Design and Intraperitoneal Injection

Vivo-Morpholinos were synthesized by Gene Tools (Philomath, Oregon, USA) and designed to inhibit mRNA splicing by targeting the junction between exon 6 and intron 6 in the zebrafish *GrnA* gene (Table 1). Intraperitoneal injections of 6.4 μg/fish Vivo-Morpholino were made in adult zebrafish. Briefly, adult fish were anesthetized with tricaine and quickly transferred to a sponge immersed in tricaine on a microscope stage. Working quickly, a needle was carefully inserted into the midline between the pelvic fins. The needle was pointed toward the rostral end of the animal and inserted closer to the pelvic girdle than the anus. PBS, GrnA vivo-MO, Ctrl-MO, or GrnA vivo-MO sense were injected to adult fish. After injection, the needle was withdrawn, and the fish was immediately transferred back to a warm-water (≈28.5 °C) tank for recovery [50].

### 4.3. Partial Hepatectomy

The adult zebrafish at 6 months old were starved for 24 h before partial hepatectomy. The fish were anesthetized with tricaine and placed on a sponge immersed in tricaine. A 5 mm 1-shaped incision was made through the ventral body skin, caudal to the ventral fin and 3 mm medial to the pectoral fin. Then, forceps and fine-spring scissors were used to resect one-half of the ventral liver lobe. After the partial hepatectomy, the fish were allowed to recover in fish water at room temperature for 2 h and then maintained at 28 °C [32]. The fish were monitored daily and analyzed at different times. Tissue samples were collected at the distal tip of the regenerating liver lobe.

### 4.4. RNA Sequencing and Analysis

Total RNA was purified from liver tissue of five fish per group of control and GrnA morphant fish at 0, 24, and 36 h after PHx. The RNA was sequenced using Illumina NextSeq sequencing (Illumina, San Diego, CA, USA). Primary analysis was performed using the Differential Gene Expression (DGE) analysis pipeline. The data are expressed as log_2_-fold changes of gene expression. The log_2_-fold changes in gene expression were determined by comparing the FPKM values of the control and GrnA-morphant fish. Genes with *p*-value ≤ 0.05 and ≥1.5-fold changes were considered significantly differentially expressed. The canonical pathway, disease and biofunction analysis, and gene network analyses were performed using Ingenuity pathway analysis (IPA) software (v1.20.4, QIAGEN IPA, Germany).

### 4.5. Cell Cycle Analysis by Flow Cytometry

The liver tissue was treated with 0.25% trypsin for 20 min and mixed well by pipetting every 5 min. FBS was added to stop the liver cell digestion, and samples were centrifuged at 1000 rpm for 10 min. Liver cells (1 × 106) were harvested and washed in cold PBS. The cells were resuspended in 200 µL cold PBS, and 200 µL cells were slowly added to cold 70% ethanol. Fixation was performed overnight in 70% ethanol at −20 °C. The cells were then centrifuged at 2000 rpm for 10 min at 4 °C, and the supernatant was carefully aspirated. Then, the cells were resuspended in PI master mix (40 µg/mL PI + 100 µg/mL RNase) and incubated at 37 °C for 30 min [51]. Finally, DNA contents of liver cells were measured by flow cytometry (Attune NxT, Invitrogen, Waltham, MA, USA) and analyzed by Flow-Jo software.

### 4.6. Quantitative Real-Time PCR

The expression levels of HGF/c-met signaling components, cell cycle regulatory genes, and cell proliferation-related genes were assessed in adult zebrafish treated with Ctrl-MO and GrnA vivo-MO at 0 to 72 h after PHx. Total RNA was extracted from each sample composed of 5 tips of remaining liver ventral lobe using TOOLSmart RNA Extractor (Biotools, New Taipei City, Taiwan), and 1 μg of total RNA was reverse transcribed using High Capacity cDNA Archive Kit (ABI). Quantitative real-time PCR analysis was performed using Power SYBR Green PCR Master Mix (ABI, Thermo Fisher Scientific, Waltham, MA, USA) according to the manufacturer’s instructions. The primers were designed on Primer3 online software (v0.4.0) (Table 2), and *EF1a* mRNA was used to normalize the relative mRNA abundance.

### 4.7. Immunohistochemistry and Western Blots

For immunohistochemistry, 10% neutral buffered formalin-fixed liver tissues were sectioned and hybridized with a PCNA antibody (PC10; Abcam, Cambridge, UK). The liver samples were collected and analyzed by Western blot to assess GrnA, HGF, and c-met expression. The lysates were hybridized with the following primary antibodies: polyclonal anti-GrnA antibody (1:1000), which was produced by immunizing BALB/c mice with 4MAPS peptide EWEDHKQKKPETQRTTTRPTG (corresponding to residues 244–264 of GrnA) (LTK Biolab Incorporation, New Taipei City, Taiwan), HGF antibody (1:1000; Aviva Systems Biology, San Diego, CA, USA), c-met antibody (1:1000, sc-10; Santa Cruz Biotechnologies, Santa Cruz, CA, USA), and GAPDH antibody (1:1000; AnaSpec, Fremont, CA, USA).

### 4.8. Imaging

All two-dimensional images were captured using Zeiss SV-11 APO Microscope and SPOT RT3 color digital camera (Zeiss, Jena, Germany) and analyzed with SPOT image software (v5.6). The liver size was quantified by measuring the fluorescence intensity and analyzed with ImageJ (Java 1.8.0_172).

### 4.9. Statistical Analysis

Fifteen fish were used per experimental group at each time point. Three independent replicates were performed. Data are presented as mean ± SD (standard deviation). Two-tailed unpaired Student’s *t*-test was used to determine the significance of differences between groups (* *p* ≤ 0.05, ** *p* ≤ 0.01, *** *p* ≤ 0.001).

## 5. Conclusions

In this study, we found that knockdown of zebrafish GrnA by IP-injection of Vivo-Morpholino led to impaired liver regeneration in partially hepatectomized zebrafish. Furthermore, inactivation of the cell cycle and suppression of cell proliferation were induced in the GrnA morphant, as indicated by relative downregulation of *ccnd1*, *ccne*, *ccna2*, and *myc.* Overall, our data suggest that GrnA positively regulates the ligand and receptor of HGF/c-met signaling genes to promote compensatory hepatocyte proliferation after partial hepatectomy.

## Figures and Tables

**Figure 1 ijms-22-11217-f001:**
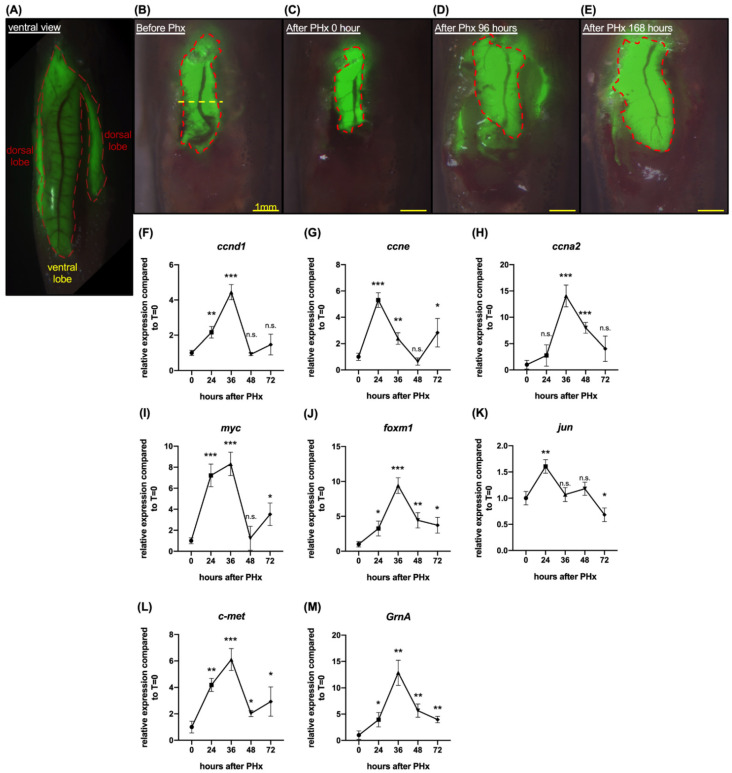
Liver regeneration was induced by one-half ventral lobe partial hepatectomy in 7 days. (**A**) Ventral view of *Tg(fabp10:EGFP)*. The green fluorescent area marked by the red dotted line is the liver of *Tg(fabp10:EGFP)*. (**B**) Schematic diagram of one-half ventral lobe partial hepatectomy (PHx). The regenerating ventral lobe is shown at 0 h (**C**), 96 h (**D**), and 168 h (**E**) after PHx. Area marked by the red dotted line is the liver ventral lobe. Yellow dotted line, the site of PHx. Scale bar: 1 mm. Gene expression was examined by quantitative PCR. Cell cycle regulatory genes included (**F**) *ccnd1*, (**G**) *ccne*, and (**H**) *ccna2*. Cell proliferation-associated genes included (**I**) *myc*, (**J**) *foxm1*, and (**K**) *jun*. (**L**) Gene expression of *c-met*. (**M**) Gene expression of *GrnA*. The relative expression levels compared to control at 0 h PHx are presented as mean ± SD. Significance was set at * *p* < 0.05, ** *p* < 0.01, *** *p* < 0.001, as determined by *t*-test; PHx, partial hepatectomy.

**Figure 2 ijms-22-11217-f002:**
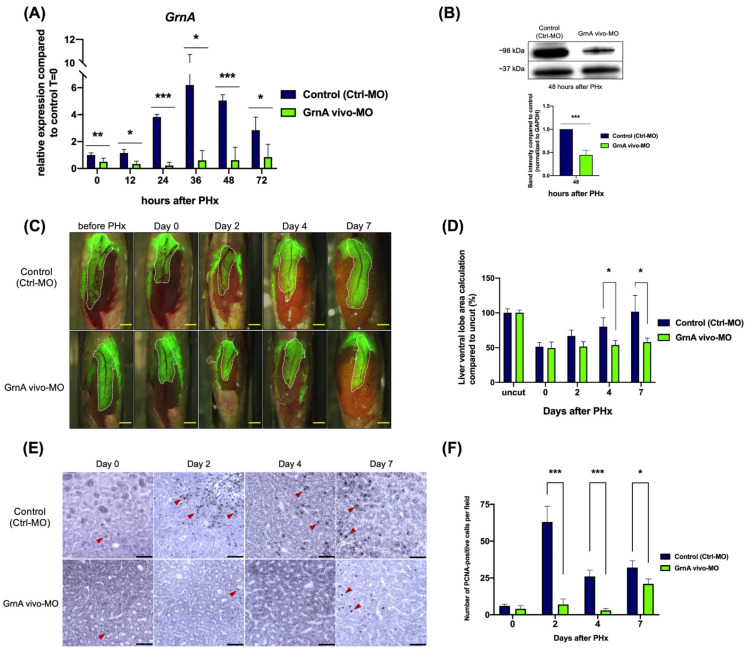
GrnA morphant exhibited impaired liver regeneration after PHx. (**A**) GrnA gene expression was examined by quantitative PCR after PHx. The relative expression levels are presented as mean ± SD, compared to control group at 0 h PHx. (**B**) Protein levels of GrnA and GAPDH were examined by Western blot analysis at 2 days after PHx. GrnA protein levels were normalized to GAPDH and then fold-change was calculated compared to control group. The values are presented as mean ± SD. (**C**) Partial hepatectomy in control (**upper**) and GrnA morphant (**lower**) was carried out on day 0. The regenerating liver ventral lobe is shown at 2, 4, and 7 days after PHx. White dotted line indicates liver ventral lobe. Red dotted line shows the site of PHx. Scale bar: 1 mm. (**D**) Liver ventral lobe area after PHx is shown in (**C**). The area was normalized to control uncut group and presented as percentage; mean ± SD. (**E**) Immunohistochemical staining of PCNA in liver ventral lobe at 0, 2, 4, and 7 days after PHx. Red arrowhead indicates the PCNA-positive cells. Scale bar: 50 μm. (**F**) Number of PCNA-positive cells was quantified after PHx, as shown in (**E**). The data are presented as mean ± SD. Significance was set at * *p* < 0.05, ** *p* < 0.01, *** *p* < 0.001, as determined by *t*-test; PHx, partial hepatectomy.

**Figure 3 ijms-22-11217-f003:**
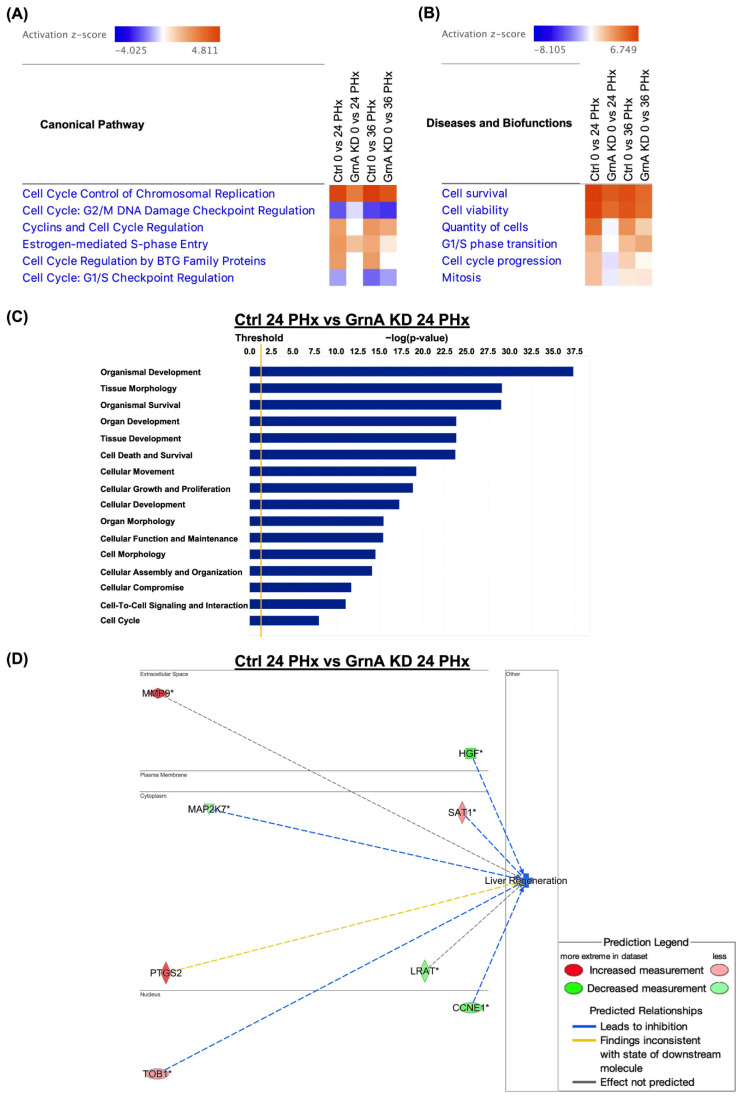
Canonical pathway, and functional and gene interaction network analyses of control and GrnA Mo-treated animals at 0, 24, and 36 h after PHx. (**A**) Heatmap shows canonical pathway analysis in comparison to control and GrnA Mo treatments at 24 and 36 h after PHx versus 0 h after PHx. (**B**) The heatmap shows disease and biofunctional analyses of control and GrnA Mo treatment at 24 and 36 h after PHx versus 0 h after PHx. Activation z-score indicates activation states of upstream transcriptional regulators: “activated” (orange) or “inhibited” (blue). Significantly activated or inhibited indicates an overlap *p*-value ≤ 0.05 and an *z*-score ≥ 1.5 (or ≤−1.5). (**C**) The top ranked diseases and biofunctions in control Mo versus GrnA Mo treatment at 24 h PHx. The threshold line represents a *p*-value of 0.05. (**D**) The gene interaction network was predicted for liver regeneration. The lines represent direct or indirect gene-to-gene interactions, and the color represents fold change. The pathway, functional analyses, and network analyses were generated using IPA. * more than one ID linked to this gene.

**Figure 4 ijms-22-11217-f004:**
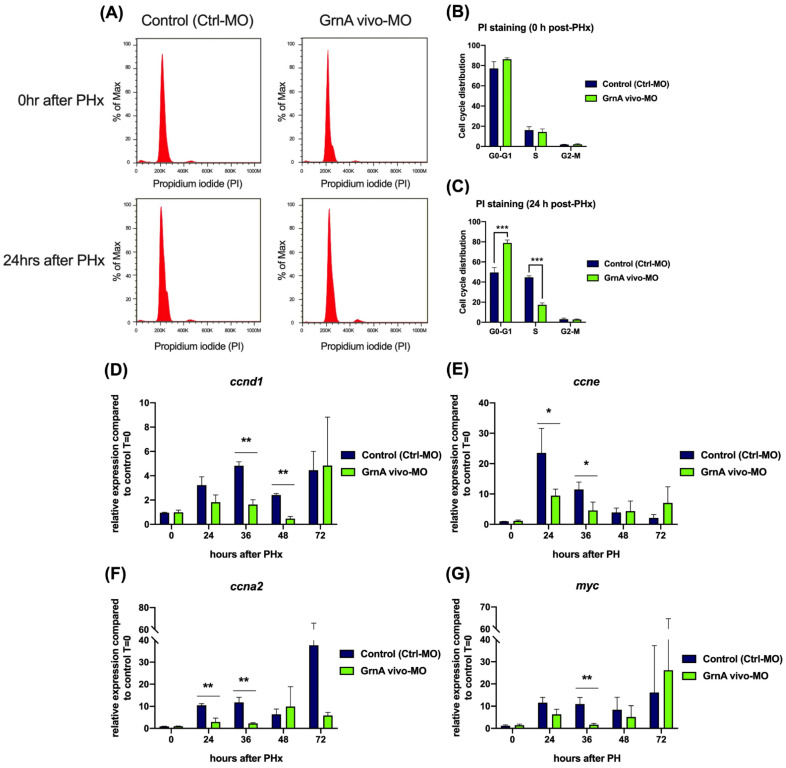
Cell cycle progression and cell proliferation were suppressed in GrnA morphants after PHx. (**A**) Cell cycle analysis of hepatectomized control and GrnA morphant zebrafish for 0 h (upper) and 24 h (lower) after PHx. PI staining was performed, and samples were analyzed using flow cytometry. The proportion of cells at each stage at 0 h (**B**) and 24 h (**C**) after PHx was analyzed by Flow-Jo software and is presented as mean ± SD. The cell cycle regulatory genes included (**D**) *ccnd1*, (**E**) *ccne*, and (**F**) *ccna2*, and (**G**) *myc*, the cell proliferation related gene, was probed by quantitative PCR. Data are shown as relative expression of indicated genes normalized to control 0 h after PHx. *EF1a* served as the internal control. Significance was set at * *p* < 0.05, ** *p* < 0.01, *** *p* < 0.001, as determined by *t*-test; PHx, partial hepatectomy.

**Figure 5 ijms-22-11217-f005:**
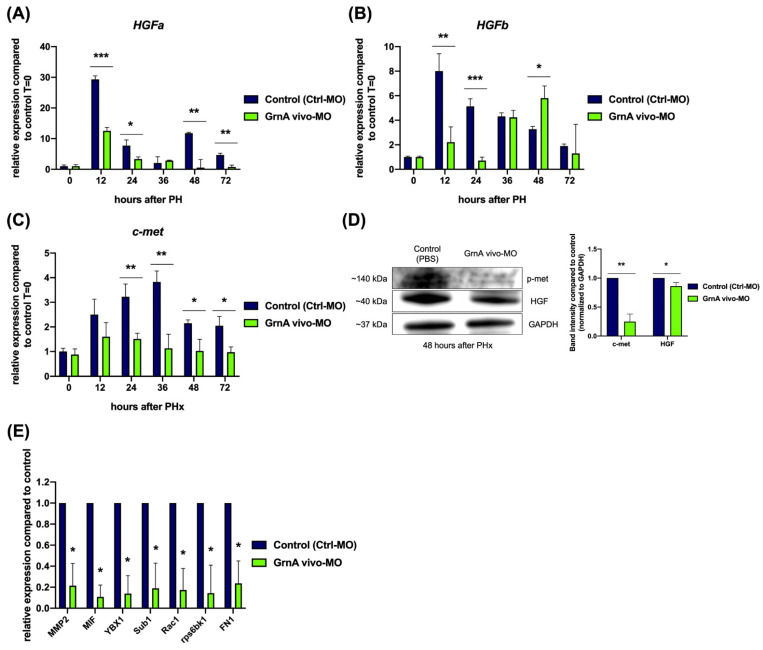
Knockdown of GrnA decreased the expression of HGF/c-met axis in hepatectomized zebrafish. Quantitative PCR analysis of zebrafish HGF paralogs, (**A**) *HGFa* and (**B**) *HGFb*, as well as the HGF-specific receptor, (**C**) *c-met*, in control and GrnA morphant zebrafish after PHx. The relative expression levels compared to control group at 0 h after PHx are presented as mean ± SD. *EF1a* served as the internal control. (**D**) Protein levels of p-met, HGF, and GAPDH were examined by Western blot analysis at 48 h after PHx. The levels of p-met and HGF proteins were normalized to GAPDH protein levels and then compared to control group. The fold change compared to controls is presented as mean ± SD. (**E**) Quantitative PCR analysis of c-met upstream signaling-related genes, including *MMP2*, *MIF*, and *YBX1*, and c-met downstream signaling-related genes, including *Sub1*, *Rac1*, *rps6kb1*, and *Fn1*. The relative expression levels compared to control group are presented as mean ± SD. *EF1a* served as the internal control. Significance was set at * *p* < 0.05, ** *p* < 0.01, *** *p* < 0.001, as determined by *t*-test; PHx, partial hepatectomy; p-met, phospho-met.

**Figure 6 ijms-22-11217-f006:**
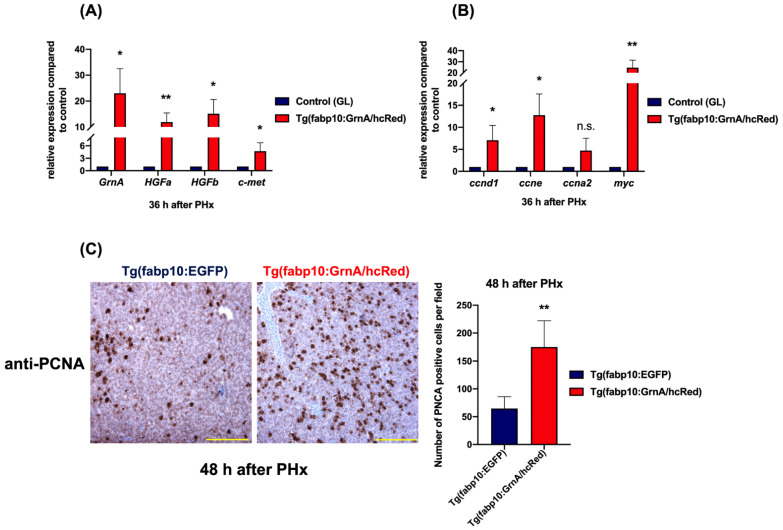
Liver-specific overexpression of GrnA positively regulated the HGF/c-met axis in hepatectomized zebrafish. (**A**) Quantitative PCR analysis of zebrafish *GrnA*, *HGFa*, *HGFb*, and *c-met* in hepatectomized zebrafish at 36 h after PHx. (**B**) The cell cycle regulatory genes (*ccnd1*, *ccne*, and *ccna2*) and cell proliferation-associated gene, *myc*, were examined by quantitative PCR. The relative expression levels compared to control group are presented as mean ± SD. *EF1a* served as the internal control. (**C**) Immunohistochemistry staining of PCNA in liver ventral lobe at 48 h after PHx. Scale bar: 50 μm. Number of PCNA-positive cells was quantified using ImageJ. The data are presented as mean ± SD. Significance was set at * *p* < 0.05, ** *p* < 0.01, as determined by *t*-test; PHx, partial hepatectomy; GL, green liver transgenic line.

**Table 1 ijms-22-11217-t001:** Vivo-Morpholinos used in this study.

Vivo-Morpholino Name	Sequence 5′ to 3′
GrnA vivo-MO	GGCTTAACTCTGCTCAATACCTTTT
Ctrl-MO	GCCTTAAATCTCCTAAATACATTTT
GrnA vivo-MO sense	AAAAGGTATTGAGCAGAGTTAAGCC

**Table 2 ijms-22-11217-t002:** Primer sequences for quantitative real-time PCR.

Primer Name	Gene	Species	Sequence 5′-3′
zf-ccnd1-F′	*ccnd1*	*Danio rerio*	CTGTGCGACAGACGTCAACT
zf-ccnd1-R′		*Danio rerio*	GGTGAGGTTCTGGGATGAGA
zf-ccne-F′	*ccne1*	*Danio rerio*	TCTTCAACCCAAAATGAGAGC
zf-ccne-R′		*Danio rerio*	CCCAAATAAAATGTTTCTCTGTGTAA
zf-ccna2-F′	*ccna2*	*Danio rerio*	CCAATAACTGAAGCCATAGCCTC
zf-ccna2-R′		*Danio rerio*	TACAAATATCTGGCTGAATCAAGC
zf-myc-F′	*myc*	*Danio rerio*	TGACTGTGGAAAAGCGACAG
zf-myc-R′		*Danio rerio*	GCTGCTGTTGATGCTGTGAT
zf-foxm1-F′	*foxm1*	*Danio rerio*	TCAGCCTGTGACCTCATCTG
zf-foxm1-R′		*Danio rerio*	AAGAGAGTGCTGTCGGGGTA
zf-jun-F′	*jun*	*Danio rerio*	AAGACCCTGAAGTCGCAAAA
zf-jun-R′		*Danio rerio*	CAAAATGTCCTTCGGCTCTC
zf-c-met-F′	*met*	*Danio rerio*	GCGCCATCAAGTCCTTAAAC
zf-c-met-R′		*Danio rerio*	GATGGCTGAAATCCTTCACG
zf-GrnA Mo-F′	*grna*	*Danio rerio*	GCATGTTCGGATGGGAAA
zf-GrnA Mo-R′		*Danio rerio*	TGTCCCGTTTCCACAAAGA
zf-HGFa-F′	*hgfa*	*Danio rerio*	GATTCATACACTCCGCACAAC
zf-HGFa-R′		*Danio rerio*	GTGGTGAAACACCAGGGAAT
zf-HGFb-F′	*hgfb*	*Danio rerio*	AAGTCATTGTTGGTGTGAGCA
zf-HGFb-R′		*Danio rerio*	AAAATGGCTGGTCGGTTATG
zf-MMP2-F′	*mmp2*	*Danio rerio*	CCCTCTGGAAGAGAGGACTGT
zf-MMP2-R′		*Danio rerio*	TTCCAGGGTACTGGCAGAAT
zf-MIF-F′	*mif*	*Danio rerio*	TGTTCGTAAAGACTCGGTTCC
zf-MIF-R′		*Danio rerio*	TCTGATCCGCGACAACCT
zf-YBX1-F′	*ybx1*	*Danio rerio*	TGTGGAGTTCGACGTGGTAG
zf-YBX1-R′		*Danio rerio*	GGGCCGGTAACATTTGCT
zf-Sub1-F′	*sub1*	*Danio rerio*	TTGTAGACCACTGCTCAACCTTAC
zf-Sub1-R′		*Danio rerio*	GCTTCTTTCTTTTAACCTTGGTTTC
zf-Rac1-F′	*rac1*	*Danio rerio*	CTCCCATCACCTACCCTCAA
zf-Rac1-R′		*Danio rerio*	GGCAGAGCACTCCAGGTACT
zf-rps6bk1-F′	*rps6bk1*	*Danio rerio*	GAGAATGTGTCCGATGACGA
zf-rps6bk1-R′		*Danio rerio*	GCCACTGCACTGGTCCAT
zf-FN1-F′	*fn1*	*Danio rerio*	AGAGGGATCACCTGCCATC
zf-FN1-R′		*Danio rerio*	GTGGTGCCCACATCAGAGA
zf-EF1a-F′	*ef1a*	*Danio rerio*	CCTCTTTCTGTTACCTGGCAAA
zf-EF1a-R′		*Danio rerio*	CTTTTCCTTTCCCATGATTGA

## Data Availability

Not applicable.

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
