# Peer review of "Progranulin A Promotes Compensatory Hepatocyte Proliferation via HGF/c-Met Signaling after Partial Hepatectomy in Zebrafish"

_ijms, 2021, doi:10.3390/ijms222011217_

Round 1

Reviewer 1 Report

In this manuscript, Chiang etal have described the role of Progranulin A in promoting liver regeneration via HGF/c-met signaling after partial hepatectomy in zebrafish. The data presented is clear. However, the overall novelty is not so high. It was previously published (cited in this work) that progranulin A-mediated MET signaling is essential for embryo liver morphogenesis in zebrafish. This work provides further insight into this signaling in liver regeneration with some mechanisms of changes in cell cycle, growth pathways, which is not totally unexpected.

Reviewer 2 Report

The manuscript is well constructed and written in fluent, easy-to-read English.
Overall, there are no faults but only a few oversights from the authors. Specifically:
In Figure 3 (C) the threshold line should be better highlighted and captions should be slightly enlarged.
On page 11, line 264, the authors state that the upregulation of HGFb is lower than CTRL at 48h post-PHx. In Figure 5 (B), the relative histogram does not match this statement. The authors should revise the text in this regard.
Overall, a good paper for which I consider the publication possible.
